# The Set1 N-terminal domain and Swd2 interact with RNA polymerase II CTD to recruit COMPASS

Hyun Jin Bae[1,6], Marion Dubarry[2,6], Jongcheol Jeon [1,3], Luis M. Soares[1,5], Catherine Dargemont[4], Jaehoon Kim [3], Vincent Geli [2✉] & Stephen Buratowski [1✉]

Methylation of histone H3 lysine 4 (H3K4) by Set1/COMPASS occurs co-transcriptionally, and is important for gene regulation. Set1/COMPASS associates with the RNA polymerase II C-terminal domain (CTD) to establish proper levels and distribution of H3K4 methylations. However, details of CTD association remain unclear. Here we report that the Set1 N-terminal region and the COMPASS subunit Swd2, which interact with each other, are both needed for efficient CTD binding in *Saccharomyces cerevisiae*. Moreover, a single point mutation in Swd2 that affects its interaction with Set1 also impairs COMPASS recruitment to chromatin and H3K4 methylation. A CTD interaction domain (CID) from the protein Nrd1 can partially substitute for the Set1 N-terminal region to restore CTD interactions and histone methylation. However, even when Set1/COMPASS is recruited via the Nrd1 CID, histone H2B ubiquitylation is still required for efficient H3K4 methylation, indicating that H2Bub acts after the initial recruitment of COMPASS to chromatin.

[1] Department of Biological Chemistry and Molecular Pharmacology, Harvard Medical School, Boston, MA 02115, USA. [2] Cancer Research Center of Marseille (CRCM), U1068 Inserm, UMR7258 CNRS, Aix Marseille University (AMU), Institut Paoli-Calmettes. Equipe labellisée Ligue contre le cancer, Marseille 13009, France. [3] Department of Biological Sciences, Korea Advanced Institute of Science and Technology, Daejeon 34141, South Korea. [4] Institute of Human Genetics, UMR 9002 CNRS, Montpellier 34396, France. [5] Present address: Foghorn Therapeutics, Cambridge, MA 02142, USA. [6] These authors contributed equally: Hyun Jin Bae, Marion Dubarry. ✉email: vincent.geli@inserm.fr; steveb@hms.harvard.edu

Eukaryotic gene transcription is regulated by posttranslational modifications of histone tails, which include phosphorylation, acetylation, ubiquitylation, and methylation[1]. Specific methylations of histone lysine residues correlate with transcription activity or repression. For instance, methylations on H3K4, K36, and K79 are enriched over genes actively transcribed by RNA polymerase II (RNApII), whereas H3K9 and H3K27 methylations are highest in transcriptionally inactive regions[2]. Methylation of H3K4 (H3K4me) has aroused particular interest[3]. In mammals, this mark can be deposited by multiple complexes (Setd1a, Setd1b, Mll1, Mll2, Mll3, and Mll4), all of which share a module comprising WDR5, RbBP5, ASH2L, and DPY-30 (called WRAD) that associates with the catalytic SET domain[4]. Each complex is endowed with additional proteins that determine their recruitment and biological functions[5].

In budding yeast, all H3K4 methylation is catalyzed by a single Set1 complex (called Set1C or COMPASS) in which Set1 acts as a scaffold for seven subunits (Bre2, Sdc1, Shg1, Spp1, Swd1, Swd2, and Swd3)[6–9]. Of these, Swd2 is the only subunit that is essential for viability. However, this requirement stems from its additional role as a component of the RNA 3′ end processing and termination complex, APT (Associated with Pta1)[10,11]. It is unclear whether Swd2 plays the same role in both APT and COMPASS. Protein interaction studies have shown that the Set1 SET domain associates with the WRAD homologs Swd1, Swd3, Bre2, and Sdc1, the N-SET domain associates with Spp1, the Set1 central region binds Shg1, and the Set1 N-terminal region contacts Swd2[7,12–15]. Swd1 also shows some interactions with the Set1 N-terminal region[14,16]. This organization of COMPASS structure was confirmed by cryo-electron microscopy[17–19].

Control of COMPASS activity involves a complex set of interactions. Remarkably, higher level H3K4 methylation by Set1 requires prior ubiquitylation on histone H2B (H2Bub)[20,21]. Spp1 contact with Swd1/Swd3 is crucial for H2Bub-dependent H3K4 methylation[16]. Deletion or depletion of individual COMPASS subunits differentially impairs Set1 stability and the pattern of H3K4 methylation along active genes[12,14,22–24]. Particularly relevant to this study, depletion of the WD40 repeat protein Swd2 strongly destabilizes Set1 and reduces H3K4 methylation[7,10,25,26]. Set1 activity is positively regulated by the Set1 double RNA recognition motif (dRRM), but inhibited by a centrally located auto-inhibitory domain[27,28]. Set1 dRMM binding to nascent RNA may affect COMPASS distribution along transcription units and subsequent deposition of the H3K4me3 mark[29]. Surprisingly, combined deletion of the N-terminal, dRMM, and central domains leads to overexpression of a truncated Set1 protein with mistargeted H3K4 methylation[16,30,31]. Finally, mutations in H3K4 lead to Set1 degradation, indicating a feedback mechanism to control enzyme levels[30].

The C-terminal domain (CTD) of Rpb1, the largest subunit of RNApII, consists of multiple Tyr1-Ser2-Pro3-Thr4-Ser5-Pro6-Ser7 (YSPTSPS) heptad repeats[32]. Chromatin immunoprecipitation (ChIP) with antibodies against individual phosphorylated CTD residues shows that phosphorylation of Serine 5 (Ser5P) peaks near the promoter, whereas Serine 2 phosphorylation (Ser2P) increases later during elongation[33]. Set1C/COMPASS co-transcriptionally associates with Ser5-phosphorylated RNApII[34] to produce a gradient of H3K4 methylation that begins at the +1 nucleosome and tails off with distance from the promoter[1,2]. This 5′ targeting is compounded over multiple transcription cycles[35], leading to the canonical peak of H3K4me3 near the promoter, followed by H3K4 dimethylation (H3K4me2) and monomethylation (H3K4me1) further downstream.

Binding of COMPASS to the RNApII CTD has been proposed to rely on the Paf1 complex[36,37]. However, direct interactions of COMPASS with either Paf1 complex or RNApII have yet to be

characterized. The recruitment of COMPASS to transcribed genes was also proposed to rely on interaction between Swd2 and H2Bub[23]. However, this model is not supported by in vitro reconstitution experiments showing that COMPASS lacking Swd2 still methylates H3K4 in an H2B ubiquitylation-dependent manner[14]. Therefore, it remains unclear how COMPASS interacts with RNApII, and whether other proteins participate in the interaction.

Here, we provide biochemical and yeast two-hybrid evidence showing that the Set1 N-terminal region and Swd2 together mediate interaction with the Rpb1 CTD. Deletion of the first 200 amino acids of Set1 results in both loss of RNApII CTD binding and reduction of H3K4me2 and H3K4me3 levels. These effects can be partially reversed by replacing this region of Set1 with the Nrd1 CTD-interacting domain (CID), which specifically binds Ser5P CTD[38]. The role of Swd2 in COMPASS recruitment is further supported by the effects of a point mutation in Swd2 that compromises its interaction with Set1, the recruitment of the complex to chromatin, and H3K4 methylation. Finally, when COMPASS is recruited by the Nrd1 "bypass" mechanism, the Paf1 complex and H2Bub are still required for H3K4 methylation, suggesting that H2Bub acts downstream of initial COMPASS recruitment to elongation complexes.

## Results

### The Set1 N-terminal domain mediates RNApII association.
Our previous analysis of Set1 N-terminal truncations found that deletion of the first 200 amino acids of Set1 strongly reduced both H3K4me2 and me3[30]. This region of Set1 interacts with the Swd2 subunit of COMPASS[14]. Interestingly, the mammalian Swd2 homolog Wdr82 preferentially binds Ser5P–CTD in vitro, and has thus been proposed to mediate Setd1A complex recruitment to elongating RNApII[15].

To test if the drop in H3K4me levels upon Set1 N-terminal truncation correlates with reduced association of COMPASS with Ser5P–CTD, a strain lacking the SET1 gene (set1Δ) was transformed with plasmids expressing either full-length Set1 or a truncation lacking 100 or 200 N-terminal amino acids (SΔ100 and SΔ200, Fig. 1a). Consistent with our previous results[30], Set1 and SΔ100 supported H3K4me3 and H3K4me2, but SΔ200 did not (Fig. 1b). To monitor binding to Ser5P–CTD, the epitope-tagged Set1 proteins were immunoprecipitated using anti-FLAG conjugated beads (α-FLAG), followed by immunoblotting for Ser5P–CTD. As shown in Fig. 1c, N-terminal deletion of Set1 (SΔ200) attenuated Set1 association with Ser5P–CTD and total RNApII. This result suggests that the N-terminal domain of Set1 (aa 1-200) that interacts with Swd2 is also important for the direct or indirect association of Set1 with RNApII.

### The N-terminal region of Set1 interacts with the Rpb1 CTD.
Evidence for a direct interaction between COMPASS and the CTD of RNApII subunit Rpb1 has been lacking. A genome-wide yeast two-hybrid (Y2H) screen was performed using full length Set1 fused to the Gal4 DNA binding domain (BD) as bait. Remarkably, a 14-CTD repeat fragment fused to the Gal4 activation domain (AD) was isolated as a robust interactor. The 14-CTD repeat construct was then tested with each of the other COMPASS subunits, but Set1 was the only subunit that showed a positive Y2H interaction (Supplementary Fig. 1a). To further map the CTD interaction, Set1 was divided into five fragments encompassing previously reported functional domains (Fig. 2a)[14]. Immunoblotting with antibodies against Gal4 BD and Set1 confirmed fusion protein expression (Supplementary Fig. 1b, c). Y2H analysis showed that full-length Set1, the F1(aa 1–236), and Set1 (1–200) fusions activated the Gal4-driven HIS3 and ADE2

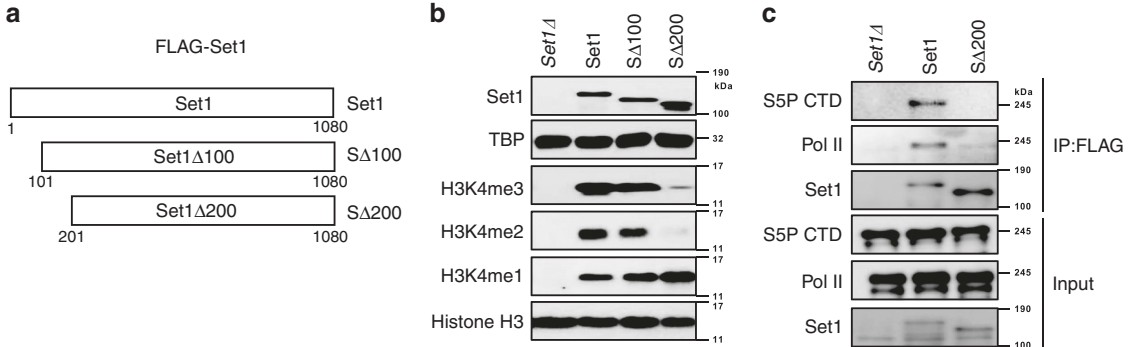

**Fig. 1 The Set1 N-terminal domain is needed for RNApII interaction and proper H3K4 methylation. a** Schematic of Set1 constructs used, with numbers below diagram indicating the amino acid residues from the wild-type proteins. A FLAG-tag at the N-terminus of all constructs is not shown. **b** Full length Set1 (Set1) and N-terminal truncated mutants (SΔ100, SΔ200) were transformed into set1Δ cells. Whole cell lysates were separated by SDS-PAGE and analyzed by immunoblotting using the indicated antibodies. Lysates from cells transformed with empty vector plasmid (set1Δ) served as a negative control. TATA-binding protein (TBP) and histone H3 were used as loading controls. **c** FLAG-tagged full-length (Set1) or Set1Δ200 (SΔ200) were expressed in set1Δ cells. Set1 proteins were immunoprecipitated with anti-FLAG beads (IP:FLAG) from whole cell lysates and analyzed by immunoblotting using antibodies for Ser5P (S5P CTD: 3E8) or total Rpb1 (POL II: 8WG16). Immunoprecipitations from cells transformed with empty vector plasmid (set1Δ) served as a negative control. Bottom panels show input samples, and upper panels show proteins bound to FLAG beads after precipitation. Source data are provided as a Source data file.

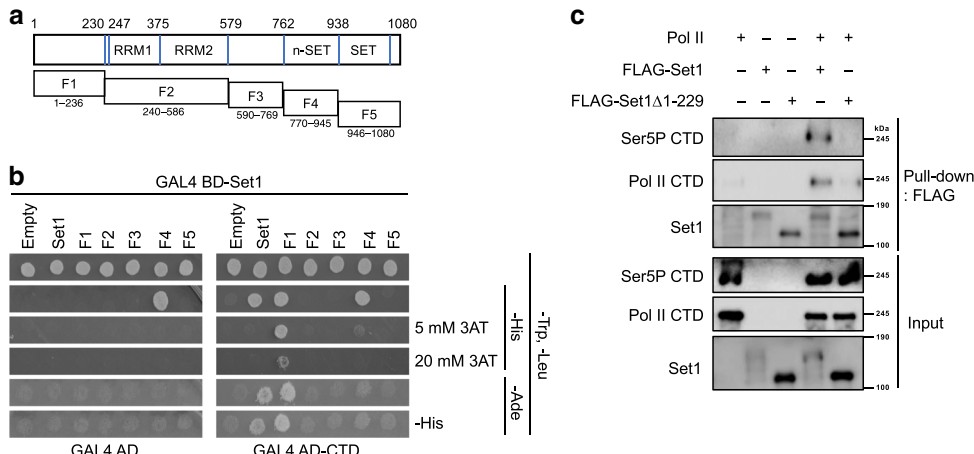

**Fig. 2 The Set1 N-terminal region binds the Rpb1 CTD. a** Schematic of Set1 fragments fused to GAL4 BD for yeast two hybrid assay. **b** GAL4 BD–Set1 fusions were co-expressed with GAL4 AD–CTD in Y2H strain PJ69-4A, which has both *HIS3* and *ADE2* reporters for Gal4 activation. Plasmid expressing only the GAL4 AD was used as a negative control. Cells were grown for 2 days on synthetic complete (SC) media plates lacking the indicated amino acids. Five or 20 mM of 3-aminotriazol (3AT) was added to media as indicated to increase *HIS3* selection stringency. Cells were also replica plated to -Ade or -Ade, -His plates to monitor the *ADE2* reporter. **c** COMPASS containing FLAG-tagged Set1 or a derivative lacking residues 1–229 was incubated with purified RNApII. Bottom panels show input samples, and upper panels show proteins bound to FLAG beads after precipitation. Source data are provided as a Source data file.

reporters, signifying interaction with RNApII CTD (Fig. 2b, Supplementary Fig. 1d). The weaker signal with the full-length Gal4 BD–Set1 fusion is due to reduced expression (Supplementary Fig. 1b, c), consistent with earlier findings that wild-type Set1 stability is tightly regulated to keep protein levels low[30]. The F4 (aa 770–945) fragment, which includes the Spp1 interacting region, also weakly activated the *HIS3* reporter, but independently of the Gal4 AD–CTD construct. Adding 3-aminotriazol (3AT) to the media selects for higher levels or *HIS3* expression, where only the F1 fragment scored as positive (Fig. 2b).

To test whether the interactions were direct, baculovirus-expressed COMPASS[16] was incubated with RNApII purified from yeast (generous gift from Dr. Naruhiko Adachi) and immunoprecipitated via the FLAG epitope on Set1. While RNApII readily bound to COMPASS containing full-length Set1, little or no interaction was seen with Set1Δ1–229 (Fig. 2c).

Therefore, we conclude that the Set1 N-terminal region is required for RNApII binding.

**Swd2 contributes to interaction between COMPASS and RNApII CTD.** Wdr82, the mammalian homolog of Swd2, can directly bind Ser5P–CTD in vitro, but also binds the N-terminal region of mammalian Setd1A[15]. We therefore considered the possibility that the amino terminal domain of Set1 interacts with the CTD only indirectly via Swd2. However, an Swd2-CTD interaction was not seen by Y2H (Supplementary Fig. 1a), arguing against this model. Furthermore, when Swd2 was directly fused to SΔ200 and introduced into cells, the fusion protein (SSΔ200) did not restore H3K4 methylation (Supplementary Fig. 2a).

One possible explanation for these results is that CTD actually binds the combination of Swd2 and Set1, either as a composite binding surface, or because one component is required to trigger

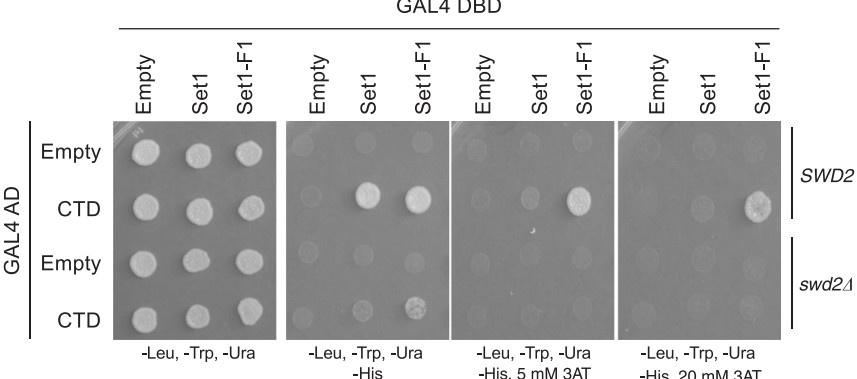

**Fig. 3 Swd2 enhances the Set1–CTD Y2H interaction.** Gal4 BD fused to full length Set1 (Set1) or amino acids 1–236 (Set1–F1, see Fig. 2) were expressed in Y2H strain PJ69-4A carrying wild-type (*SWD2*) or deletion (*swd2Δ*) alleles of *SWD2*. They were combined with a Gal4 AD fusion to the Rpb1 CTD (CTD) or the parent vector (Empty). Cells were spotted on the indicated selective plates. Source data are provided as a Source data file.

a CTD-binding conformation in the other. To test this possibility, the Y2H interaction between the Gal4 AD–CTD fusion and Gal4 BD fused to full-length Set1 or segment F1 was compared in isogenic reporter strains that were either wild-type *SWD2* or *swd2Δ*. The lethality of *swd2Δ* was suppressed, as previously described, by expression of a fragment from the termination factor Sen1 (amino acids 1890–2092, annotated as Sen1(202)[25]). The control *SWD2* strain was also transformed with the same Sen1 construct to maintain isogeneity. Immunoblotting once again showed that Gal4 fused to Set1 F1 is expressed at higher levels than the full-length Set1 fusion, but also that *swd2Δ* does not affect amounts of either protein (Supplementary Fig. 2b). In the Y2H assay, activation of the Gal4-responsive *HIS3* reporter was severely reduced, but not abolished by *swd2Δ* (Fig. 3). In agreement, *swd2Δ* strongly reduces H3K4me3 in cells with wild-type Set1, but not in cells with the Set1 N-terminal truncation (SΔ200) (Supplementary Fig. 2c). These results suggest that Swd2, while not absolutely required, strongly promotes proper interaction between the Set1 and CTD fusion proteins.

Additional deletions more precisely mapped Swd2 binding to aa 124–229 of Set1[16]. In agreement with this finding, deleting amino acids 200–210 from the Set1-F1 fragment (F1Δ200–210) weakened its yeast two-hybrid interaction with both Swd2 and CTD Gal4 AD fusions (Supplementary Fig. 3a). In the context of full length Set1, Δ200–210 only slightly affected Set1 protein levels, but H3K4me3 levels (Supplementary Fig. 3b) and ChIP of Set1 to the 5′ region of *PMA1* gene were reduced (Supplementary Fig. 3c). These results further show that Swd2 contributes to the COMPASS–RNApII interaction.

**The Nrd1 CID can substitute for the Set1 N-terminal domain.** If the primary role of the Set1 N-terminal region and associated Swd2 subunit is to recruit COMPASS to the elongation complex via CTD binding, it should be possible to replace this domain with another Ser5P-binding protein. The CID from Nrd1 targets the snoRNA termination machinery to 5′ ends by direct binding to Ser5P–CTD[38]. Accordingly, we created a fusion where the CID replaces the N-terminal 200aa of Set1 (Fig. 4a). Strikingly, the Nrd1(CID)–SΔ200 fusion (NSΔ200) partially rescued both bulk H3K4me2 and me3 (Fig. 4b), as well as the ability to co-precipitate Ser5-phosphorylated RNApII (Fig. 4c). To prove that restoration of H3K4 methylation patterns by the Nrd1 CID fusion is mediated by CTD binding, we created two Set1 fusions with Nrd1 CID point mutants (D70R or I130R) known to disrupt the CTD interaction[38]. Co-immunoprecipitation experiments confirmed that D70RSΔ200 and I130RSΔ200 fusion proteins were expressed normally, but had lost the ability to stably bind RNApII

(Fig. 4c). Neither mutant restored H3K4 methylation relative to SΔ200, as expected if CTD binding is essential for proper COMPASS targeting (Fig. 4b).

To determine if this CID-mediated rescue of H3K4 methylation reflects normal or aberrant distribution along genes, H3K4me2 and me3 patterns were analyzed genome-wide by ChIP and high-throughput sequencing (ChIP-Seq). Spiked-in *S. pombe* chromatin was used as an internal control. Both individual gene heat maps (Fig. 4d, e, left and middle panels) and averaged "meta-gene" anchor plots (Supplementary Fig. 4a, b) show that SΔ200 significantly reduced H3K4me3 peaks. These effects can also be seen in representative Mochiview[39] genome browser tracks for individual genes (Supplementary Fig. 4c, d), as well as in heat maps quantitating the differences between SΔ200 and Set1 FL (Supplementary Fig. 4e, f, left panels). Relative to SΔ200, the NSΔ200 fusion increased both promoter–proximal H3K4me3 and downstream H3K4me2 (Fig. 4d, Supplementary Fig. 4). Thus, the NSΔ200 methylation patterns are intermediate between wild-type Set1 and SΔ200 cells, consistent with improved recruitment of COMPASS by the Nrd1 CID. We have previously shown that trimethylation occurs over multiple rounds of transcription[35], so in mutants with reduced COMPASS occupancy or activity, promoter–proximal nucleosomes maximally attain H3K4me2, while downstream nucleosomes that would normally have H3K4me2 only reach H3K4me1, making it appear that the H3K4 methylation gradient has shifted upstream[35].

Because the NSΔ200 fusion replaces the Set1 Swd2-binding region with the Nrd1 CID, its interaction with RNApII is predicted to be independent of Swd2. To test this, we compared the behaviors of wild-type Set1, SΔ200, and NSΔ200 in *SWD2/set1Δ* versus *swd2Δ/set1Δ* cells. As previously seen[25,30], full-length Set1 is degraded in cells lacking Swd2. In contrast, levels of SΔ200 or NSΔ200 were not reduced in *swd2Δ* cells (Fig. 4f). H3K4me3, assayed either in bulk (Fig. 4f) or by ChIP (Supplementary Fig. 2c), was strongly stimulated by Swd2 in *SET1* cells, but not in SΔ200 or NSΔ200. Deletion of *SWD2* also strongly diminished interaction between RNApII and wild-type Set1. In contrast, co-immunoprecipitation of RNApII with NSΔ200, while lower than wild-type Set1, was independent of Swd2 (Fig. 4g). These observations further suggest that Swd2 and the Set1 N-terminal region cooperate in COMPASS targeting.

**The Swd2 WD40 domain is important for COMPASS recruitment.** To characterize the role of Swd2 WD40 domain in COMPASS–RNApII interactions, a point mutant was created at phenylalanine 250, located at the center of the WD40 domain at the tip of propeller blade 6 (Supplementary Fig. 5a, b). Protein levels of

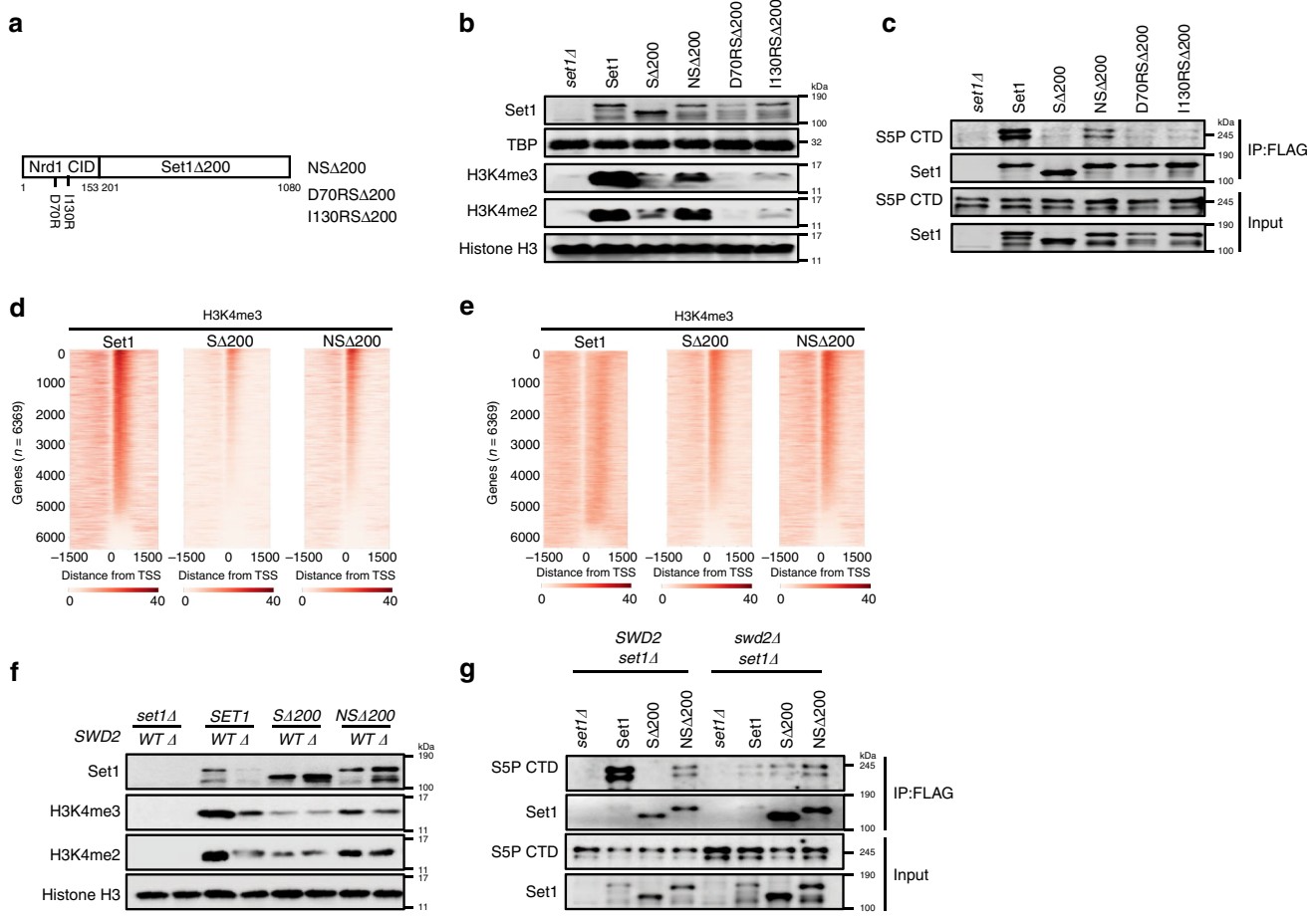

**Fig. 4 Nrd1 CID fusion to Set1Δ200 partially restores CTD-binding and H3K4 methylation. a** Schematic diagram of the NSΔ200 fusion protein consisting of the Nrd1 CID (amino acids 1–153) and Set1Δ200. A FLAG-tag at the N-terminus is not shown. The positions of two CID mutants that disrupt CTD binding are also shown. **b, c** FLAG-tagged full-length Set1 (Set1), Set1Δ200 (SΔ200), or Nrd1 CID fusions (NSΔ200, D70RSΔ200, and I130RSΔ200) were transformed into set1Δ cells. Whole-cell lysates (**b**) and proteins immunoprecipitated using anti-FLAG beads (**c**) were analyzed by immunoblotting using indicated antibodies. Cells transformed with empty plasmid (set1Δ) served as a negative control. **d, e** ChIP-Seq heat maps of H3K4me3 (**d**) and H3K4me2 (**e**) from strains carrying full-length Set1 (Set1), Set1Δ200 (SΔ200), or Nrd1CID–Set1Δ200 fusion (NSΔ200). After normalizing to an internal spiked-in sample of *S. pombe* chromatin, SPMR values were mapped for individual RNApII transcriptional units and ordered according to H3K4me3 values in Set1 cells. H3K4me levels within 1500 bps from transcription start site (TSS) are shown. ChIP-seq reads from two technical repeats were averaged and plotted. **f, g** FLAG-tagged full-length Set1 (Set1), Set1Δ200 (SΔ200), and Nrd1CID–Set1Δ200 (NSΔ200) were transformed into set1Δ or set1Δswd2Δ cells. **f** Whole-cell lysates or **g** anti-FLAG immunoprecipitates were separated by SDS-PAGE and analyzed by immunoblotting using the indicated antibodies. Histone H3 was used as a loading control. Source data are provided as a Source data file.

the F250A mutant were close to those of wild-type Swd2. In contrast, H3K4me2 and me3 levels were much lower in the Swd2 mutant, indicating partial disruption of COMPASS activity (Fig. 5a). As we previously reported for other mutants with reduced H3K4 methylation, Set1 levels were also reduced (Fig. 5a). Immunoprecipitation of COMPASS via either Spp1 or Swd3, whose levels were unaffected in the mutant, or Swd2 itself, also showed that Swd2 F250A was defective for association with COMPASS (Supplementary Fig. 5c). To rule out that loss of interaction with Swd2 F250A was simply an indirect effect of Set1 degradation, we expressed HA-tagged WT or F250A Swd2 in the presence of FLAG-tagged Set1 and untagged WT Swd2, thereby maintaining a supply of functional and stable COMPASS methylation activity. Immunoprecipitation of Set1 confirms that the F250A mutation strongly reduces Swd2 association with Set1 (Fig. 5b).

The yeast two-hybrid interaction between Set1 and the CTD was compared in cells containing WT or F250A Swd2. Similar to the Swd2 deletion (Fig. 3), the reduced interaction in the mutant background suggests that the Swd2 WD40 domain facilitates Set1/COMPASS binding to RNApII (Fig. 5c). Immunoprecipitation

with an antibody for CTD Ser5P pulled down less Spp1 and Swd2 in the F250A mutant, further supporting this conclusion (Supplementary Fig. 5d). Interestingly, ChIP-qPCR shows that Swd2 F250A is normally localized (Supplementary Fig. 5g), although this may reflect Swd2 in APT rather than free Swd2. However, Swd2 F250A diminishes enrichment of Spp1 and H3K4me3 at the *PMA1* 5′ region (Supplementary Fig. 5e, f), while Pol II occupancy is normal (Supplementary Fig. 5h). Supporting these in vivo results, reconstituted recombinant Set1/COMPASS complex incorporated lower levels of F250A than wild-type Swd2 (Fig. 5d) and exhibited reduced H3K4 methyltransferase activity on nucleosomes in vitro (Fig. 5e). Although these in vitro effects are less severe than in vivo, this is likely because H3K4 methylation defects in vivo are further amplified by Set1 degradation. Altogether, the F250A data implicate the Swd2 WD40 domain in COMPASS integrity and recruitment to early elongation complexes.

**H2Bub is required for H3K4 methylation independently of Swd2.** Ubiquitylation of histone H2B on lysine 123 (H2Bub)

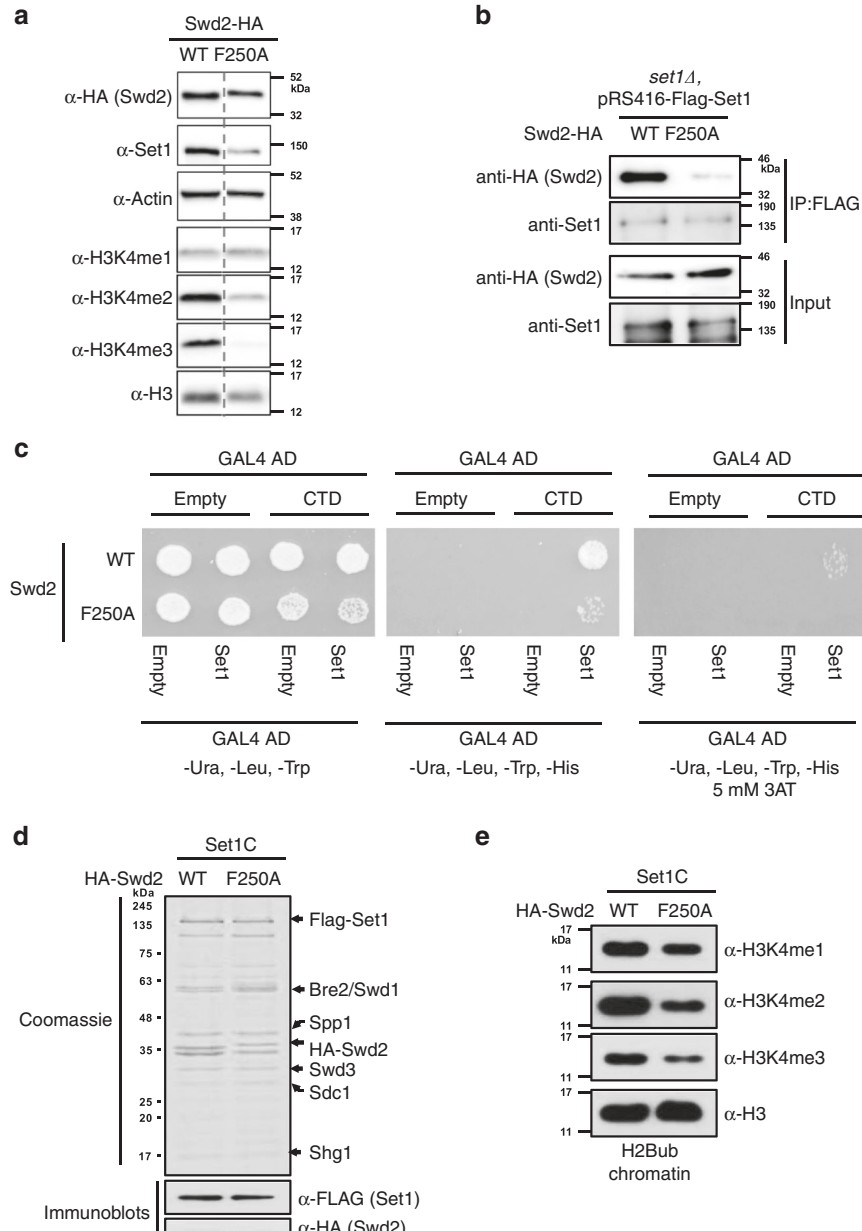

**Fig. 5 The Swd2 WD40 domain is critical for COMPASS function. a** Swd2–HA or Swd2–F250A–HA were expressed in an *swd2Δ* deletion strain. Protein levels were analyzed by immunoblotting using the indicated antibodies. Actin and Histone H3 were used as loading controls. Note that samples were run on the same blot, but intervening lanes were removed (dashed line, full blots appear in Source data). **b** Plasmid-encoded Swd2–HA and Swd2–F250A–HA were expressed in FLAG-Set1 containing cells that also express the endogenous untagged *SWD2* gene in the chromosome. Proteins were immunoprecipitated with anti-FLAG beads (IP:FLAG) and then immunoblotted for Set1 and Swd2-HA. **c** The Y2H interaction between the Gal4BD–Set1 and Gal4AD–CTD fusions was tested in strain PJ69-4A expressing wild-type (WT) or mutant (F250A) Swd2. The empty Gal4 vectors served as negative controls. Activation of the Gal4-activated HIS3 reporter was tested on plates lacking histidine, with the addition of 5 mM 3AT in the last panel for additional stringency. **d** Analysis of recombinant baculovirus-expressed Set1/COMPASS made with WT Swd2 or the Swd2–F250A mutant. SDS-PAGE/Coomassie blue staining shows total protein, while immunoblotting with anti-FLAG and anti-HA show Set1 and Swd2 levels, respectively. **e** H2Bub chromatin was subjected to in vitro histone methyltransferase assays[16] with Swd2 WT and Swd2-F250A mutant-containing complexes shown in panel (**d**). H3K4 methylation status was monitored by immunoblotting with indicated antibodies. Histone H3 was used as an internal loading control. Source data are provided as a Source data file.

facilitates higher level H3K4 methylations by COMPASS[22]. H2Bub is targeted to transcribed regions by the Paf1 elongation factor complex (Paf1C)[37,40], which recruits the H2Bub ubiquitination complex Rad6-Bre1[20,21,41]. Deleting any of these components causes loss of H2Bub and reduced H3K4me2 and me3[20,21,37]. It has been proposed that Swd2 mediates recognition of H2Bub near 5′ ends of genes to facilitate binding of

COMPASS[22,23]. If the primary role of H2Bub was mediated via Swd2, the Swd2-independent fusion of Set1 to the Nrd1 CID might also bypass the requirement for H2Bub.

As expected, immunoblotting of cell extracts showed that H3K4me2 and me3 were reduced below the level of detection in *paf1Δ* or *rtf1Δ* cells (Fig. 6a), as well as in *bre1Δ* cells (Supplementary Fig. 6a). Replacing the first 200 amino acids of

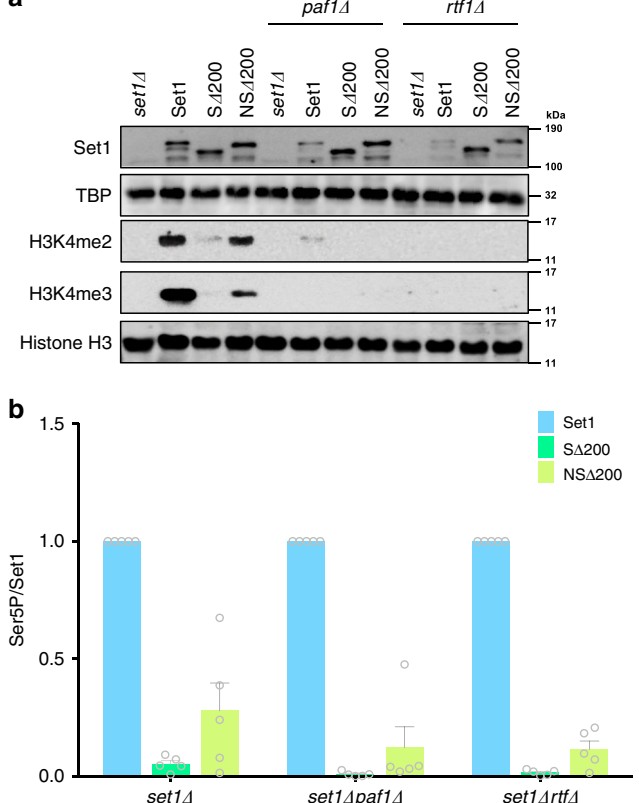

**Fig. 6 Nrd1CID–Set1Δ200 fusion does not bypass the role of Paf1C and H2Bub in H3K4me. a** Wild-type Set1 (Set1), Set1Δ200 (SΔ200), and Nrd1CID–Set1Δ200 (NSΔ200) were transformed into *set1Δ*, *set1Δpaf1Δ*, or *set1Δrtf1Δ* cells. Protein levels and histone methylations in whole-cell extracts were analyzed by immunoblotting. TBP and Histone H3 were used as loading controls. Parallel experiments in *set1Δbre1Δ* cells showed the same loss of H3K4 methylation (Supplemental Fig. 6a). **b** Set1 association with RNApII was quantitated by immunoprecipitating Set1 using α-FLAG beads and probing for Rpb1 CTD Ser5P and Set1. Set1 and Ser5P signal intensities from each immunoblot were quantitated using ImageJ, and Ser5P signals were normalized to those for Set1. Circles show individual experimental values, and error bars show s.e.m. ($n = 5$). Values from wild type Set1 transformants were normalized to 1.0 for each experiment. Source data are provided as a Source data file.

Set1 with the Nrd1 CID partially restored the ability of Set1 to co-immunoprecipitate CTD–Ser5P, even in the Paf1C or Bre1 mutants (Fig. 6b, Supplementary Fig. 6b), but did not restore methylation (Fig. 6a, Supplementary Fig. 6a). The lack of methylation rescue was further confirmed by ChIP-qPCR on the *YEF3* gene (Supplementary Fig. 7). These results echo our earlier demonstration that H3K4 methylation by a fusion of Set1 to the Rpb4 polymerase subunit is also still dependent upon H2Bub[35]. Therefore, artificial recruitment of COMPASS does not bypass the requirement for H2Bub, indicating this modification promotes H3K4 methylation independently of COMPASS association with the RNApII elongation complex. This conclusion has been supported by several structures of nucleosome-bound COMPASS that were published while this paper was in review (see Discussion below)[42–44].

## Discussion

The details of how H3K4 methylation is coordinated with active transcription are still not completely understood. Although Set1/

COMPASS targeting is known to involve CTD Ser5 phosphorylation[34], it has been unclear whether COMPASS directly binds the CTD, whether RNApII domains other than CTD are involved, and which subunits of COMPASS mediate the interaction. Here, we present evidence that the N-terminal region of Set1 and Swd2 cooperatively interact with RNApII–CTD to promote proper recruitment of COMPASS to transcription elongation complexes at 5′ ends of genes. This model is based on the following observations:

i. Deleting the first 200 amino acids of Set1 (SΔ200) strongly reduced H3K4me3 and the ability to co-precipitate RNApII.
ii. Swd2, which interacts with amino acids 124–229 of Set1[16,19], is needed for strong CTD binding by Set1.
iii. Deleting residues 200–210 of Set1 abrogates Y2H interaction with both the Rpb1 CTD and Swd2.
iv. An Swd2 WD40 domain point mutation impairs its interaction with both Set1 and RNApII, recruitment of COMPASS to chromatin, and H3K4 methylation activity.
v. Replacing amino acids 1–200 of Set1 with the Nrd1 CID (NSΔ200) partially restores H3K4me and Ser5P–CTD binding, showing that CTD binding is a major function of this region.

The Nrd1 CID fusion only partially substitutes for the Set1 N-terminal domain, which might reflect lower CTD affinity. More likely, Set1(1–200) and Swd2 have additional functions, beyond CTD interaction, in regulating COMPASS activity, as indicated by physical interaction studies[16], protein crosslinking, and low-resolution cryo-EM structural analysis[19]. It appears that the Swd2–Set1 N-terminal module may fold over the catalytic body of COMPASS to contact other subunits and more C-terminal Set1 regions. An interesting speculative model is that CTD binding moves the Swd2–Set1 N-terminal module, relieving autoinhibition by making the Set1 active site more accessible.

Several facts led us to initially suspect that Swd2 would be the primary contact point with the CTD. Yeast Swd2 is also a component of the APT complex, which functions in the yeast Nrd1–Nab3–Sen1 (NNS) termination pathway for short non-polyadenylated transcripts. Like COMPASS, NNS termination is also directed to early stage RNAPII elongation complexes marked by Ser5-phosphorylated CTD[38], and this common CTD preference could be related to their shared subunit. Indeed, genetic evidence suggests APT and COMPASS compete for a common interacting protein, perhaps RNApII itself, through Swd2[26]. Interestingly, *S. pombe* has two Swd2 homologs, each specific for one of the two complexes[45]. In addition, the mammalian Swd2 homolog Wdr82 helps recruit the Setd1A to chromatin and can directly bind to CTD Ser5P in vitro[15]. Echoing our results, this in vitro binding was stimulated by the N-terminal region of mammalian Set1. Despite our expectations, we could not detect a CTD–Swd2 interaction in the Y2H assay, nor did direct fusion of Swd2 to Set1Δ200 rescue H3K4 methylation. Therefore, yeast Swd2 is apparently not sufficient for CTD interaction. Altogether, these results suggest that the N-terminal region of Set1 and Swd2 may form a composite binding site for tethering COMPASS to the RNApII CTD. It may be that both proteins make CTD contacts, or alternatively, that one triggers a conformation that renders the other competent for CTD binding.

In contrast to Set1Δ200, Set1 derivatives further C-terminally truncated to amino acids 700[35] or 762[27,31] can tri-methylate H3K4. However, the resulting H3K4me3 is mislocalized, spreading from the 5′ region into the gene bodies of active genes. This inaccurate distribution of H3K4me3 still requires Spp1, as removal of the Spp1 interaction site by Set1 truncation to 780–1080 leads to loss of H3K4me3[18,27,31,46,47]. Thus, other

modes of COMPASS recruitment to chromatin must exist in the absence of the Set1 region 1–762, likely via direct interactions of the COMPASS catalytic domain with the nucleosome[42–44], or perhaps through transient interactions between Swd2 and COMPASS subunits Spp1 and Swd1[19]. In the context of full-length Set1, these alternative modes of recruitment may be inhibited by the central autoinhibitory domain of Set1[27].

In addition to CTD binding, co-transcriptional methylation by COMPASS requires H2Bub. Paf1C associated with transcription elongation complexes directs Bre1/Rad6 ubiquitin ligase to nucleosomes, and H2Bub in turn stimulates H3K4 methylation[36,37,48–50]. Although the Nrd1 CID fusion to Set1 can restore CTD binding, it does not bypass the requirement for H2Bub (Fig. 6, Supplemental Fig. 6). The H2Bub requirement also remained upon fusion of Set1 to RNApII subunit Rpb4[35] or strong overexpression of the hyperactive Set1 N-terminal truncations[30]. After normalizing for reduced Set1 protein levels in the absence of H3K4 methylation[30,35], the ability of Set1 to co-precipitate RNApII was also unaffected by loss of Paf1C or Rad6-Bre1 ubiquitination complex (Fig. 6b). Therefore, H2Bub must act at a step after initial recruitment of COMPASS to the RNApII elongation complex. Indeed, several structures published while this paper was under review show how ubiquitin contacts and allosterically activates COMPASS[42–44]. While H2B-linked ubiquitin does not affect COMPASS affinity for nucleosomes[14], its interactions with Set1 and other subunits rearrange the methyl-transferase catalytic site into an active conformation.

A model thus emerges in which nucleosome binding, H2Bub sensing, and methyltransferase activity map to the C-terminal N-SET/SET region of Set1 and associated COMPASS subunits, while an N-terminal domain combines with Swd2 to create a CTD-targeting module. Future biochemical and structural studies will eventually reveal how these two modules interact, and how they are regulated by an intervening region of Set1 linked to auto-inhibition and COMPASS degradation. The C-terminal N-SET/SET domains and WRAD subunits are the most conserved among the Set1/MLL family, consistent with their common H3K4 methyltransferase activities. Our experiments suggest that the non-conserved N-terminal domains are likely to target individual family members to different genomic locations through distinct protein interactions.

## Methods

**Strains, plasmids, and primers**. Yeast strains used in this study are listed in Supplementary Table 1. Plasmids used for fusion protein analysis and yeast two hybrid assay are described in Supplementary Tables 2 and 3, respectively. DNA encoding the CID from Nrd1 (amino acids 1–153) was amplified from yeast genomic DNA and inserted into the indicated Set1 constructs using isothermal assembly[51]. Primers used for cloning or ChIP assays are listed in Supplementary Table 4.

**Antibodies**. The following antibodies were used in this study: anti-H3K4me1 (Millipore Sigma 07-436, Burlington, MA, 1:2000), anti-H3K4me2 (Millipore Sigma 07-030, 1:1000), anti-H3K4me3 (Millipore Sigma 07-473 or 04-745, 1:2000), anti-H3 (Abcam 1791, Cambridge, UK, 1:3000), rat monoclonal anti-Ser2P CTD (3E10, Dirk Eick, 1:1000), rat monoclonal anti-Ser5P CTD (3E8, Dirk Eick, 1:3000), mouse monoclonal anti-CTD for total Rpb1 (8WG16, Buratowski lab, 1:1000), anti-Set1 (Santa Cruz, sc-101858, 1:1000), anti-TBP polyclonal antiserum (Buratowski lab, 1:3000), anti-Myc (MMS-150R-500, Covance, 1:2000), anti-β-Actin (Abcam 8224, 1:2000), anti-HA (3F10, Roche and 12CA5, 1:2000), anti-Gal4 DBD (sc-577, Santa Cruz, 1:1000).

**Co-immunoprecipitation and immunoblotting**. Whole-cell lysate was prepared from 100 ml of yeast cultures grown at 30 °C until $OD_{600}$ reached to 1.0. Harvested cells were resuspended in 1 ml lysis buffer (50 mM HEPES-KOH [pH 7.5], 150 mM NaCl, 0.1% Triton X-100, 10% Glycerol, 1 mM DTT) supplemented with protease and phosphatase inhibitors (1 µg/ml leupeptin, 1 µg/ml aprotinin, 1 µg/ml pep-statin A, 1 µg/ml antipain, 1 mM NaF, 1 mM $Na_3VO_4$, and 1 mM PMSF). Protein concentrations were determined by Coomassie Protein Assay (Bio-Rad, Hercules, CA). For FLAG immunoprecipitation, 3 mg of lysates were incubated at 4 °C

overnight with 10 µl of anti-DYKDDDDK L5 agarose beads (Biolegend, San Diego, CA) that had been preblocked with 0.1% bovine serum albumen. Beads were washed twice for 5 min with 1 ml of lysis buffer.

To assay binding with purified proteins (Fig. 2c), 5 µg of purified yeast RNApII (generously provided by N. Adachi) and 100 ng of Flag-tagged Set1 complex purified from insect cells were incubated overnight at 4 °C in 500 µl of incubation buffer (50 mM HEPES, pH7.6, 100 mM potassium acetate, 10 mM magnesium acetate, 1 mM EDTA, 10% glycerol and 0.1% NP-40). The mixtures were immunoprecipitated for 2 h at at 4 °C using 10 µl of anti-DYKDDDDK L5 agarose beads that had been preblocked with 0.1% bovine serum albumen. Beads were washed three times for 5 min with 1 ml of incubation buffer.

Immunoprecipitated proteins were eluted by boiling with 50 µl of sodium dodecyl sulfate (SDS) sample buffer. For immunoblotting, 25–50 µg of whole cell lysates or 10 µl of IP eluates were resolved by SDS polyacrylamide gel electrophoresis (PAGE) and transferred onto polyvinylidene fluoride membranes (Millipore, Billerica, MA). The membrane was blocked with blocking buffer (5% powdered skim milk in tris-buffered saline, 0.1% Tween-20) and probed with indicated antibodies above. Chemiluminescence signals were detected by using SuperSignal West Pico or Femto substrate (ThermoFisher Scientific, Waltham, MA) and visualized using the LAS 3000 image analyzer (Fuji Photo Film, Tokyo, Japan).

**Chromatin Immunoprecipitation (ChIP)**. Chromatin samples were prepared as previously described[30]. Cells were crosslinked with 1% formaldehyde for 20 min, followed by 5 min quenching with 3 M glycine. Cells were lysed by vortexing with glass beads (30 × 30 s, with cooling between cycles, total 20) in FA lysis buffer (50 mM HEPES-KOH [pH7.5], 150 mM NaCl, 1 mM EDTA, 1% triton X-100, 0.1% sodium deoxycholate, 0.5% SDS, supplemented with protease inhibitors). Cell debris was removed by microcentrifugation, and the chromatin sheared to ~200 bp using a Misonix 3000 cup horn sonicator. Insoluble material was removed by microcentrifugation for 10 min at 14,000 rpm at 4 °C, and final protein concentration determined using Coomassie Protein Assay (Bio-Rad).

For immunoprecipitation, 500 µg of chromatin was incubated with 0.5 µl of anti-H3K4me2, or anti-H3K4me3 and 10 µl of Protein G-Sepharose beads at 4 °C overnight in FA lysis buffer with SDS reduced to 0.1%. For ChIP-Seq, chromatin from *S. pombe* was added at 10% relative to *Saccharomyces cerevisiae* chromatin as a "spike-in" control. Precipitates were washed with same buffer containing 275 mM NaCl (H3K4me2) or 500 mM NaCl (H3K4me3). The beads were washed with Wash Buffer (10 mM Tris-HCl [pH 8.0], 0.25 M LiCl, 1 mM EDTA, 0.5% NP-40, 0.5% Na-Deoxycholate), and TE (10 mM Tris-HCl [pH8.0], 1 mM EDTA) buffer. Precipitated materials were eluted with buffer containing 50 mM Tris-HCl [pH7.5], 10 mM EDTA and 1% SDS by incubating at 65 °C for 20 min. Subsequent decrosslinking was performed at 42 °C for 2 h and at 65 °C for overnight with 0.8 mg/ml of pronase (VWR, Radnor, PA). DNAs were phenol–chloroform extracted followed by ethanol-precipitation for further analysis.

**Preparation of ChIP-Seq Libraries**. Sequencing libraries were prepared using the following procedure, as previously described[35]. The concentration of immuno-precipitated DNA was measured using Qubit dsDNA HS Assay kit (Thermo Fisher Scientific). Barcoded sequencing libraries were generated from 1 ng of immuno-precipitated DNA[52]. Briefly, immunoprecipitated DNA was end repaired using T4 DNA polymerase, T4 PNK, and DNA polymerase I Large (Klenow) fragment (New England Biolabs, Ipswich, MA). A single adenosine was added to the 3′ end of fragments using Klenow (3′ to 5′ exo minus, New England Biolabs) and then adapters containing multiplexing barcodes were ligated using T4 Quick DNA ligase (New England Biolabs). Adapter ligated DNA fragments between 200 and 500 bp were gel purified and PCR amplified with 16 cycles using Phusion DNA poly-merase (ThermoFisher Scientific). Quality of libraries was examined using Agilent 2100 Bioanalyzer (Agilent Santa Clara, CA). Equimolar amount of each library was mixed and 50 bp single-end sequenced in High Output (Standard) v3 Illumina HiSeq 2000 (Harvard Bauer Center Core Facility, Cambridge, MA).

**Bioinformatics analysis**. Analysis of ChIP-seq data was performed using the pipeline described in Soares et al.[35]. Sequence reads were demultiplexed using SABRE (https://github.com/najoshi/sabre.git) allowing for one mismatch in the barcode. For *S. pombe* spike-in normalization, demultiplexed reads were first aligned to the *S. pombe* genome (version ASM294v2.31). The unaligned reads were subsequently aligned to the *S. cerevisiae* genome (version R64-1-1). Only sequence reads that could be exclusively assigned to each genome were considered for total number of reads. Normalization factors were calculated by first calculating the proportion of reads of *S. pombe* versus total reads in input reads, and dividing the value by the square root (to account for the sequence tags per million reads (SPMR) normalization) of the proportion of reads of *S. pombe* vs. total reads in each immunoprecipitation reads. All alignment was performed using BOWTIE 1.1.1[53] excluding the first base and multi-aligned reads.

Conversion of alignment files was performed using SAMTOOLS 1.2[54]. Using MACS2.1.0[55] function, pileup tracks were calculated and duplicate reads were removed. Subsequently, tags were extended to 150, and values were normalized to SPMR. Final coverage outputs were converted to high density wig files and

analyzed using custom Python3.4 scripts (http://www.sciencelint.org/, https://github.com/LuisSoares/Manuscript). The GEO accession number for the sequence data reported in this paper is GSE138281.

**Yeast two-hybrid assay (Y2H).** The *SET1* open reading frame was cloned as a bait into vector pB66 (N-GAL4 BD-bait-C fusion). Initial interaction screening of a *S.cerevisiae* genomic library was performed by Hybrigenics, SA, Paris (https://www.hybrigenics-services.com/contents/our-services/interaction-discovery/ultimate-y2h-2). For subsequent Y2H validation and further analysis, *S.cerevisiae* reporter strains PJ69-4A or CG1945 were used. Gal4 DBD and Gal4 AD plasmids used in Y2H are listed in Supplementary Table 3. Interactions were scored 2 or 3 days after spotting on SC media plates lacking the appropriate amino acids for reporter gene activation (Gal4-dependent UAS upstream of *HIS3* or *ADE2*). For more stringent selection, 5 or 20 mM 3-aminotriazole (3-AT), a competitive inhibitor of the *HIS3*-gene product, was added to media as indicated.

**Quantitative PCR (qPCR) analysis.** For individual gene analysis, quantitative PCR reactions of DNAs from ChIPs (see above) were done with a BioRad CFX384 using the following parameters: 5 min at 95 °C, 40 cycles of 15 s at 95 °C, 15 s at 50 °C, and 40 s at 72 °C, followed by 10 min at 95 °C. Oligonucleotides used for PCR reactions are listed in Supplementary Table 4.

**COMPASS purification and histone methyltransferase assay.** Purification of Set1 complex, H2Bub chromatin assembly, and in vitro methyltransferase assay were performed as described previously[16,56]. Briefly, cDNAs amplified from yeast genomic DNA were subcloned into pFASTBAC1 (ThermoFisher Scientific) with or without an epitope tag and baculoviruses were generated according to the manufacturer's instruction. Set1 complexes containing Swd2 wild-type or F250A mutant were reconstituted from Sf9 cells which were infected with combinations of baculoviruses. Proteins/complexes were affinity purified using M2 agarose (Millipore Sigma, Burlington, MA). H2Bub chromatin assembly was performed using the recombinant ACF/NAP1 system. For recombinant chromatin methyltransferase assays, reactions containing 350 ng (based on histone amount) recombinant chromatin (35 μl, assembled as above), purified Set1 complexes and 100 μM SAM (S-adenosyl methionine, New England Biolabs) were adjusted to 40 μl with HEG buffer (25 mM HEPES [pH 7.6], 0.1 mM EDTA and 10% glycerol) and incubated at 30 °C for 1 h. Proteins were resolved by SDS-PAGE and subjected to immunoblotting.

**Statistics and reproducibility.** All western blots were repeated at least three times and representative images were shown in this paper. Unpaired *t* test was used for statistical analysis. For ChIP-Seq, mean values from two biological replicates are represented.

**Reporting summary.** Further information on research design is available in the Nature Research Reporting Summary linked to this article.

## Data availability

The ChIP-seq data sets generated and analyzed in Fig. 4 and Supplementary Fig. 4 are available in the GEO repository, accession number GSE138281 (https://www.ncbi.nlm.nih.gov/geo/query/acc.cgi?acc=GSE138281). All other relevant data supporting the key findings of this study are available within the article and its Supplementary Information files or from the corresponding authors upon reasonable request. Uncropped images for gels and blots in all figures are provided in the Source data file. A reporting summary for this Article is available as a Supplementary Information file.

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

## Acknowledgements

H.B. was supported by a postdoctoral fellowship from the Basic Science Research Program of the National Research Foundation of Korea (NRF), funded by the Ministry of Education (2015R1A6A3A03017730). This work was supported by grants from the National Research Foundation of Korea (NRF-2019R1A2C2090830) to J.K., from "Ligue Nationale Contre le Cancer" (LNCC) (Equipe labellisée) to M.D. and V.G., and grants GM046498 and GM056663 from the U.S. National Institutes of Health to S.B. We thank D. Eick (CIPS, Munich) for CTD antibodies, S. Briggs (Purdue) for the parent FLAG-Set1 construct, C. Gwizdek (Dargemont lab) for creating the Swd2 F250A mutant, and N. Adachi (SBRC, KEK, Japan) for purified RNApII.

## Author contributions

H.B. and M.D. performed all the experiments, with the exception of COMPASS reconstitution experiments performed by J.J. and J.K., and computational analyses of genomic data by L.S. and H.B. C.D. identified and initially characterized the *SWD2* F250A mutant. H.B., M.D., V.G., and S.B. participated in experimental conception and design, and wrote the paper with input from all other authors.

## Competing interests

The authors declare no competing interests.
