## [Peer Review File · Nature Communications]

Reviewers' comments:

Reviewer #1 (Remarks to the Author):

In this report Bae et al. study the recruitment of Set1C to the Ser5 phosphorylated CTD of RNA Pol II (S5P CTD). Set1C is recruited during transcription to methylate K4 of histone H3 (H3K4me_{1,2,3}). Methylation depends on histone H2 ubiquitination (H2BK123ub), which has been proposed to be required for the recruitment of Set1C. The authors show that deletion of the first 200aa of Set1 leads to loss of H3K4me₂ and H3K4me₃, together with loss of the interaction with S5P CTD and Swd2, another component of Set1C. The interaction of the N-ter of Set1 with S5P CTD is sensitive to the integrity of Swd2, which led the authors to propose that Set1C interacts with S5P CTD via a combined surface made of Set1 Nter and Swd2. The recruitment of Set1 to the CTD can be partially recapitulated by a chimera in which the Nter of Set1 is replaced with the S5P CTD binding domain of Nrd1, which restores some of the phenotypes of the deletion and loses the Swd2 dependency. Interestingly, although the recruitment of the chimera to the CTD is partially restored in an Swd2 independent manner, the H2BK123ub requirement for methylation is not bypassed, which challenges the model of H2BK123ub-dependent recruitment.

This study sheds light on the function of the Set1C complex and the mechanism of its recruitment. The data presented are of high quality and generally support the model proposed. The results are of interest to a wide audience of readers, and therefore would be appropriate for publication in Nature Communication. Below are a few points that should be addressed before acceptance.

- Fig. 1c : the Pol II signal obtained after IP of Set1 variants is unclear and similar in all lanes. This observation is in apparent contrast with the strong difference in the signal obtained with the S5P CTD antibody. If the pol II antibody detects all Pol II forms, one possibility is that SΔ200 interacts preferentially with another phosphovariant (e.g. Ser2P), which should be assessed. In any case this – possibly apparent – discrepancy should be discussed.

- Figure 4b-c: the CID mutants D70R and I130R are not described and discussed in the text, which should be corrected as these mutants provide important controls.

- Figure 4d-e and supplementary figure 4: the distribution of H3K4me₃ in the chimera NSΔ200 is indeed intermediate between the wild type and SΔ200, indicating the chimera partially suppresses the me₃ phenotype. However, the distribution of H3K4me₂ is strongly altered in NSΔ200 and resembles more the distribution of SΔ200 than the wild type (i.e. it is clearly 5'-shifted relative to the wt, fig. S4b). Why is it so ? Is this a consequence of the failure to efficiently convert me₂ to me₃ ? Also, why there is less me₂ in the body of genes in NSΔ200 relative to wt ? The authors should discuss this and provide plausible explanations.

- In Fig 4g the restoration of the S5P CTD interaction in NSΔ200 is not very convincing (contrary to what shown in Fig 4c). A quantification of the signals would help.

- Fig. 5B: Set1 is strongly destabilized by F250A in Swd2 (see input in 5b and 5a) and it is therefore difficult to conclude that this mutation affects the interaction of Swd2 with Set1. Loss of interaction could be inferred because of Set1 destabilization (although this is quite indirect), but not really from the IP.

- It is not completely clear what are the effects of the F250A mutation in Swd2. In Fig. 5e, showing the in vitro reconstitution of the complex, there is no apparent decrease of Spp1 and a modest decrease of Swd2, which is in contrast to the strong effect reported in vivo (Fig 5b). I do not think that it can be concluded that Swd2 WD40 domain is implicated in COMPASS integrity and recruitment to early elongation (lines 234-235). It is possible that the interaction with Set1 is

affected and that all the phenotypes observed derive from the destabilization of Set1.

Reviewer #2 (Remarks to the Author):

In this manuscript, Buratowski and colleagues address the important question of how the Set1/COMPASS complex is recruited to RNA Pol II. Previous work revealed a requirement for CTD serine 5 phosphorylation (S5P) in recruiting COMPASS to active genes. Here, the authors use yeast two-hybrid, co-immunoprecipitation, and ChIP experiments to demonstrate that the N-terminal region of Set1 (amino acids 1-200) is necessary and sufficient for interacting with the CTD in vivo. They also show that the Set1-CTD interaction requires a functional copy of Swd2, which is a member of both the COMPASS and APT complexes. Unlike the situation with the mammalian counterpart of Swd2, Wdr82, the authors did not detect an interaction between Swd2 and the CTD. Instead, their data support a coordinate role for the Set1 N-terminus and Swd2 in binding to CTD in its S5P state. Additional support for this model comes from a clever protein fusion experiment in which the authors replace the N-terminus of Set1 with the CTD-interacting domain of a transcription termination factor, Nrd1. The resulting fusion protein is only partially functional in rescuing H3 K4me3 levels in cells, arguing either that the Nrd1 domain has weaker affinity for the CTD than the Set1 N-terminal domain or that the Set1 N-terminal domain, along with Swd2, has additional functions. Finally, the authors show that recruiting Set1 through the Nrd1 CID is insufficient to bypass the requirement for H2B ubiquitination in H3K4me3. In summary, the main contribution of this study is the demonstration that association of COMPASS with RNA Pol II S5P requires an interaction between the N-terminal region of Set1 and the CTD, which is mediated by Swd2. Previous work on mammalian Set1 and Swd2/Wdr82 showed binding of Swd2/Wdr82 to the CTD and stimulation by the N-terminal region of Set1. Overall, the work here is convincing and expands upon the process in yeast. The results argue that Swd2 and the N-terminal region of Set1 are important for recruitment of COMPASS to RNA Pol II and driving H3K4me patterns.

Specific points:

1. At several points in the paper, the authors comment on the unknown role of H2B ubiquitination in H3K4me by COMPASS and speculate on the roles of various COMPASS subunits in this process. However, a paper published in Nature in mid-September (Nature 573: 45) has provided a structural explanation for MLL recognition of a ubiquitinated nucleosome and needs to be included in the authors' interpretations. (A similar structure is now available for a yeast COMPASS catalytic module bound to a ubiquitinated nucleosome in bioRxiv.) These structures don't take away from the authors' work on CTD binding by Swd2 and the Set1 N-terminal domain, which are not in the nucleosome-bound structures, but should be discussed.
2. The Y2H experiments show Set1 amino acids 1-200 are sufficient for an interaction with a CTD prey protein. The authors seem to suggest these results show a direct interaction between Set1 and the CTD. Because Y2H interactions may involve bridging proteins between the bait and prey, a direct interaction cannot be concluded from the Y2H or co-IP experiments in this paper. The question of whether COMPASS is directly binding to the CTD remains unanswered. With the ability to reconstitute COMPASS with purified components, it's curious that the authors have not tested binding of COMPASS to RNA Pol II or CTD peptides in vitro? They seem to be well set up to do it and could even include some interesting mutants in the experiment.
3. Figure 1c: Why is the signal for Pol II essentially equivalent in all the immunoprecipitated samples? The enriched signal for S5P CTD in the FL Set1 IP is clear, but total Pol II levels (which antibody?) in the IP samples appear independent of Set1 or its form. The data seem inconsistent.
4. Statistical analysis is needed for ChIP assays and Figure 6b.

5. Figure 4b: I might have missed it, but I don't think the mutants in lanes 5 and 6 are discussed in the Results.
6. Can the authors propose an explanation for the 5' shift in H3K4me2 patterns in the Set1 Δ 200 constructs?
7. The Swd2-F250A mutant reduces Spp1-Myc occupancy on PMA1. What is the effect on Set1 occupancy? How disruptive is this mutation to complex integrity in cells? This is important because the authors would like to argue that this mutation is having specific effects on COMPASS recruitment to RNA Pol II. If the entire COMPASS complex is disrupted, a reduction in H3K4me3 would not be as interesting. The co-IPs in Figure 5b show greatly reduced association of Spp1 and Set1 with the Swd2-F250A mutant. What about other subunits? It doesn't seem relevant to compare the co-IPs with the in vitro reconstitution. In the reconstituted complex the Swd2 mutation does not seem to affect Set1 and Spp1 incorporation in the complex. Also, the effects of the F250A substitution on H3K4me3 in vitro are subtle compared to effects in vivo. Fitting with the model, this could be due to the requirement for Swd2 in recruiting RNA Pol II to genes in vivo, but this was not clearly stated in that section of the Results. Overall, more caution is needed in interpreting this mutant unless the authors have additional data supporting its specificity.
8. Does the Swd2-F250A mutation disrupt the Y2H interaction between Set1 and the CTD?
9. Figure S1b: Molecular weight markers are needed.
10. Line 162: "isogeneicity" is misspelled.
11. Figure S2b: There are many bands in the Set1 lanes. On what basis was the band for full length Set1 assigned? Was the blot probed with anti-Set1 antibody?
12. Figure S2c: Can the authors explain the H3K4me3 enrichment in the 3' UTR? Is there a noncoding transcript in this region?

Reviewer #3 (Remarks to the Author):

Review comments:

The study by Bae et. al. focused on characterization of the direct interaction between Pol II-CTD and subunits of the ySET1 complex. Despite the general assumption that ySET1 complex is directly recruited by Pol II to target genes, the exact molecular mechanism is not clear. The manuscript clearly demonstrated that N-terminal of ySET1 and Swd2 cooperatively interact with Pol II-CTD and this interaction is important for H3 K4 methylation in yeast. Furthermore, it showed that this n-SET/Swd2/Pol-II CTD regulation is independent of PAF1 and H2BK120ub and thus convincingly argued against a previous study that Swd2 functions through Bre1/Rad6 in H3K4me regulation. Most of the experiments were performed at high quality and with well-designed controls. There are only several weaknesses that need to be addressed before publication.

1. It is interesting that S Δ 200 and NS Δ 200 shift H3K4me2 to promoter proximal regions. Better explanation is warranted. Does this mean that these ySET1 mutants no longer interact with Pol II? This would argue against the proposed function by NS Δ 200.
2. Figure 4f, why NS Δ 200 had two bands?
3. Figure 4g, the pull-down assay seemed to suggest a much weaker interaction between CTD and NS Δ 200. However, rescue for H3K4me2/3 seemed much better. What is the reason for

discrepancy?

4. Figure 5 discussed a Swd2 mutant F250A, which was defective for Spp1 interaction. Since Spp1 is important for H3K4me regulation and indirectly interact with Pol II, it is not clear what exactly the function of this mutant. Is it possible that this mutant functions through spp1 instead of the Set1/Swd2/CTD axis? This has to be better clarified. If this mutant is indeed function through Spp1, what is the rationale to include it here?

5. A recent cryo-EM study for the ySET1-NCP complex by Hsu et al., is available in bioRxiv. It elegantly demonstrates how H2BK120ub regulates H3K4me by removing the autoinhibitory ARM loop in n-SET. This should be cited and discussed. Given this new information, figure 6 is probably not necessary or at least be discussed in the context of the ySET1-NCP structure.

Detailed Responses to Reviewers' Comments:

Authors' responses are in italics.

Reviewer #1 (Remarks to the Author):

In this report Bae et al. study the recruitment of Set1C to the Ser5 phosphorylated CTD of RNA Pol II (S5P CTD). Set1C is recruited during transcription to methylate K4 of histone H3 (H3K4me1,2,3). Methylation depends on histone H2 ubiquitination (H2BK123ub), which has been proposed to be required for the recruitment of Set1C. The authors show that deletion of the first 200aa of Set1 leads to loss of H3K4me2 and H3K4me3, together with loss of the interaction with S5P CTD and Swd2, another component of Set1C. The interaction of the N-ter of Set1 with S5P CTD is sensitive to the integrity of Swd2, which led the authors to propose that Set1C interacts with S5P CTD via a combined surface made of Set1 Nter and Swd2. The recruitment of Set1 to the CTD can be partially recapitulated by a chimera in which the Nter of Set1 is replaced with the S5P CTD binding domain of Nrd1, which restores some of the phenotypes of the deletion and loses the Swd2 dependency. Interestingly, although the recruitment of the chimera to the CTD is partially restored in an Swd2 independent manner, the H2BK123ub requirement for methylation is not bypassed, which challenges the model of H2BK123ub-dependent recruitment.

This study sheds light on the function of the Set1C complex and the mechanism of its recruitment. The data presented are of high quality and generally support the model proposed. The results are of interest to a wide audience of readers, and therefore would be appropriate for publication in Nature Communication. Below are a few points that should be addressed before acceptance.

We really appreciate the reviewer's positive assessment and very helpful comments below.

- Fig. 1c : the Pol II signal obtained after IP of Set1 variants is unclear and similar in all lanes. This observation is in apparent contrast with the strong difference in the signal obtained with the S5P CTD antibody. If the pol II antibody detects all Pol II forms, one possibility is that Δ 200 interacts preferentially with another phosphovariant (e.g. Ser2P), which should be assessed. In any case this – possibly apparent – discrepancy should be discussed.

Our original IP had very high non-specific binding of total RNAPII to the commercially purchased anti-FLAG beads (which is why we saw RNAPII, but not Ser5P, signal even in lane 1 of the old Fig 1C, where the cells completely lack Set1). We have now repeated this experiment using beads that have been more thoroughly pre-blocked with BSA. With the reduction in background, we see that total RNAPII also shows preferential binding to full-length Set1 as expected (New Fig 1C, see below). For simplicity, we did not include a strip for CTD Ser2P here, as we and several other labs (for example, Ng et al, Cell 2003) have previously shown that Set1 specifically IPs Ser5P and not Ser2P.

c

- Figure 4b-c: the CID mutants D70R and I130R are not described and discussed in the text, which should be corrected as these mutants provide important controls.

Thanks for pointing this out. The sentences describing these important controls accidentally got deleted during our editing process. They have been restored.

- Figure 4d-e and supplementary figure 4: the distribution of H3K4me3 in the chimera NS Δ 200 is indeed intermediate between the wild type and S Δ 200, indicating the chimera partially suppresses the me3 phenotype. However, the distribution of H3K4me2 is strongly altered in NS Δ 200 and resembles more the distribution of S Δ 200 than the wild type (i.e. it is clearly 5'-shifted relative to the wt, fig. S4b). Why is it so? Is this a consequence of the failure to efficiently convert me2 to me3? Also, why there is less me2 in the body of genes in NS Δ 200 relative to wt? The authors should discuss this and provide plausible explanations.

We have tried to better explain in the text (p9) why this is exactly the result we expect, based on our model for H3K4 methylation in Soares et al (2017). There we proposed that H3K4me3 at promoters is proportional to the total time COMPASS is near the nucleosome over multiple rounds of transcription. While Ser5P tethering can lead to me1 or maybe me2 during a single round of transcription, me2 and me3 largely result from the fact that nucleosomes are recycled during transcription so that later rounds of transcription convert me1 to me2 and me2 to me3. One other factor that needs to be considered is that recycled nucleosomes are eventually passed from the 3' to 5' ends of genes (called "treadmilling" in the 2011 Radman-Livaja paper), which would lead to more me2 and me3 near the promoter even in the absence of COMPASS-Ser5P interaction. Because the NS Δ 200 only partially restores Ser5P binding, we see more K4me3 compared to S Δ 200, but less than in WT. Similarly, the K4me2 that would normally be seen further downstream is only able to get to the K4me1 stage. Because of space limitations, we don't completely restate the conclusions and model from Soares et al, but hope readers will go back and read that paper as well if they still don't understand the results.

- In Fig 4g the restoration of the S5P CTD interaction in NS Δ 200 is not very convincing (contrary to what shown in Fig 4c). A quantification of the signals would help.

We agree that the earlier figure presented the results in a manner that made it hard to see the rescue. We re-ran the exact same samples, but loaded the lanes in a different order, separating SWD2 from the swd2 deletions. We believe the new panel 4g makes interpretation much easier. The quantifications below show that the level of partial rescue by NS Δ 200 is slightly less in 4g, but still clear when placed next to S Δ 200.

Fig.4c and g

- Fig. 5B: Set1 is strongly destabilized by F250A in Swd2 (see input in 5b and 5a) and it is therefore difficult to conclude that this mutation affects the interaction of Swd2 with Set1. Loss of interaction could be inferred because of Set1 destabilization (although this is quite indirect), but not really from the IP.

As we reported in Soares et al., 2014, there is a feedback mechanism by which any mutation that reduces H3K4 methylation (even the H3K4A mutation) results in Set1 destabilization. This makes it very difficult to deconvolute whether COMPASS defects cause methylation changes or vice versa. However, we showed in that paper that normally degraded Set1 mutants were stabilized in trans by expressing a second (untagged) functional copy of Set1, so we tried the same trick with Swd2. We expressed HA-tagged versions of WT or F250A Swd2 in the presence of an untagged Swd2, which allows us to monitor Swd2/Set1 co-IP under conditions where functional COMPASS remains present. We IP the FLAG-tagged Set1 and then monitor with anti-HA for the tagged WT or F250A Swd2. As can be seen in the new Fig 5B (also shown here), WT and mutant Swd2 are expressed equally, but only WT is efficiently co-IPd with Set1. This validates our conclusion that the F250A mutation disrupts the interaction of Swd2 with Set1.

- It is not completely clear what are the effects of the F250A mutation in Swd2. In Fig. 5e, showing the in vitro reconstitution of the complex, there is no apparent decrease of Spp1 and a modest decrease of Swd2, which is in contrast to the strong effect reported in vivo (Fig 5b). I do not think that it can be concluded that Swd2 WD40 domain is implicated in COMPASS integrity and recruitment to early elongation (lines 234-235). It is possible that the interaction with Set1 is affected and that all the phenotypes observed derive from the destabilization of Set1.

The new COMPASS structures show that Spp1 interacts with the COMPASS catalytic core (WRAD) independently of Swd2, so the reviewer is correct that loss of Swd2-Spp1 coIP seen in Fig 5b is indirectly due to the loss of Swd2-Set1 interaction. We apologize if we inadvertently implied otherwise. Therefore, we don't expect a reduction of Spp1 in the reconstituted complex. However, we completely agree that the reconstitution of COMPASS show a much milder defect in Swd2 F250A association than we see in vivo. We have shown that yeast cells have one or more "quality control" mechanisms for tuning COMPASS levels through degradation of defective or excess COMPASS (again, see Soares 2014). Therefore, we suspect that the partial association defect in the recombinant COMPASS system gets amplified in vivo because the F250A complexes are defective and trigger further COMPASS degradation. We have added a sentence with this suggestion. We believe that the new Fig 5B shows that F250A disrupts the interaction with COMPASS, independently (or at least upstream) of the Set1 feedback degradation.

Reviewer #2 (Remarks to the Author):

In this manuscript, Buratowski and colleagues address the important question of how the Set1/COMPASS complex is recruited to RNA Pol II. Previous work revealed a requirement for CTD serine 5 phosphorylation (S5P) in recruiting COMPASS to active genes. Here, the authors use yeast two-hybrid, co-immunoprecipitation, and ChIP experiments to demonstrate that the N-terminal region of Set1 (amino acids 1-200) is necessary and sufficient for interacting with the CTD in vivo. They also show that the Set1-CTD interaction requires a functional copy of Swd2, which is a member of both the COMPASS and APT complexes. Unlike the situation with the mammalian counterpart of Swd2, Wdr82, the authors did not detect an interaction between Swd2 and the CTD. Instead, their data support a coordinate role for the Set1 N-terminus and Swd2 in binding to CTD in its S5P state. Additional support for this model comes from a clever protein fusion experiment in which the authors replace the N-terminus of Set1 with the CTD-interacting domain of a transcription termination factor, Nrd1. The resulting fusion protein is only partially functional in rescuing H3 K4me3 levels in cells, arguing either that the Nrd1 domain has weaker affinity for the CTD than the Set1 N-terminal domain or that the Set1 N-terminal domain, along with Swd2, has additional functions. Finally, the authors show that recruiting Set1 through the Nrd1 CID is insufficient to bypass the requirement for H2B ubiquitination in H3K4me3. In summary, the main contribution of this study is the demonstration that association of COMPASS with RNA Pol II S5P requires an interaction between the N-terminal region of Set1 and the CTD, which is mediated by Swd2. Previous work on mammalian Set1 and Swd2/Wdr82 showed binding of Swd2/Wdr82 to the CTD and stimulation by the N-terminal region of Set1. Overall, the work here is convincing and expands upon the process in yeast. The results argue that Swd2 and the N-terminal region of Set1 are important for recruitment of COMPASS to RNA Pol II and driving H3K4me patterns.

We are grateful for the reviewer's positive evaluation and very helpful comments.

Specific points:

1. At several points in the paper, the authors comment on the unknown role of H2B ubiquitination in H3K4me by

COMPASS and speculate on the roles of various COMPASS subunits in this process. However, a paper published in Nature in mid-September (Nature 573: 45) has provided a structural explanation for MLL recognition of a ubiquitinated nucleosome and needs to be included in the authors' interpretations. (A similar structure is now available for a yeast COMPASS catalytic module bound to a ubiquitinated nucleosome in bioRxiv.) These structures don't take away from the authors' work on CTD binding by Swd2 and the Set1 N-terminal domain, which are not in the nucleosome-bound structures, but should be discussed.

While our paper was being reviewed, we were excited to see the multiple structure papers now showing how H2Bub interacts with the catalytic core subunits of COMPASS. We now incorporate this new information into our discussion. While these structures by themselves do not rule out additional interactions of H2Bub with Swd2, they are very consistent with our conclusions that bypassing Swd2 does not alleviate the requirement for H2Bub and PAF complex.

2. The Y2H experiments show Set1 amino acids 1-200 are sufficient for an interaction with a CTD prey protein. The authors seem to suggest these results show a direct interaction between Set1 and the CTD. Because Y2H interactions may involve bridging proteins between the bait and prey, a direct interaction cannot be concluded from the Y2H or co-IP experiments in this paper. The question of whether COMPASS is directly binding to the CTD remains unanswered. With the ability to reconstitute COMPASS with purified components, it's curious that the authors have not tested binding of COMPASS to RNA Pol II or CTD peptides in vitro? They seem to be well set up to do it and could even include some interesting mutants in the experiment.

This is an excellent suggestion, and an experiment we certainly had on our list. The new Fig 2c shows that we can see binding of purified RNAPII to recombinant COMPASS containing full length Set1, but not with a deletion removing the first 229 amino acids. We believe this experiment makes it very unlikely that any additional bridging proteins are required.

3. Figure 1c: Why is the signal for Pol II essentially equivalent in all the immunoprecipitated samples? The enriched signal for S5P CTD in the FL Set1 IP is clear, but total Pol II levels (which antibody?) in the IP samples appear independent of Set1 or its form. The data seem inconsistent.

This point was also raised by Reviewer 1. Our original IP had very high background binding of total RNAPII (8WG16) to the beads (which is why we got signal even in lane 1, where the cells completely lack Set1). We have now repeated this experiment using beads that have been pre-blocked with BSA. With the reduction in background, we see that total RNAPII also shows preferential binding to full-length Set1 as expected (New Fig 1C, also shown here).

c

4. Statistical analysis is needed for ChIP assays and Figure 6b.

The ChIP assays (now in Sup Figs 5e-h) and IPs (Fig 6b, now based on 5 replicates) both show s.e.m. error bars, and P-values below 0.05 are marked. Calculated P-values for Fig 6b fall between 0.05 and 0.27 (not shown in the actual figure, but see graph below). However, because the values being plotted here are normalized ratios (Ser5P/Set1, setting the WT situation to 1.0) rather than single variables, they incorporate the variability in both the Ser5P (numerator) and Set1 (denominator) signals. It's therefore not clear that P-values are an appropriate test of statistical significance here, with most of the variation actually coming from the very small numbers in the "negative control" reaction with SA200. Given the current backlash against over-interpreting strict P-value cutoffs, we prefer to

leave the figure as is. In five repeats of this experiment, the trends are always the same, so we are very confident in our conclusion that the Nrd1 CID partially restores interactions with RNAPII, even in the absence of H2Bub.

5. Figure 4b: I might have missed it, but I don't think the mutants in lanes 5 and 6 are discussed in the Results.

Thanks for pointing this out. The sentences describing these important controls accidentally got deleted during our editing process. They have been restored.

6. Can the authors propose an explanation for the 5' shift in H3K4me2 patterns in the Set1Δ200 constructs?

This point was also raised by Reviewer 1. We have tried to better explain in the text (p9) why this is exactly the result we expect, based on our model for H3K4 methylation in Soares et al (2017). We proposed that H3K4me3 at promoters is proportional to the total time COMPASS is near the nucleosome over multiple rounds of transcription. While Ser5P tethering can lead to me1 or maybe me2 during a single round of transcription, me2 and me3 largely result from the fact that nucleosomes are recycled during transcription so that later rounds of transcription convert me1 to me2 and me2 to me3. One other factor that needs to be considered is that recycled nucleosomes are eventually passed from the 3' to 5' ends of genes (called "treadmilling" in the 2011 Radman-Livaja paper), which would lead to more me2 and me3 near the promoter even in the absence of COMPASS-Ser5P interaction. Because the NSΔ200 only partially restores Ser5P binding, we see more K4me3 compared to SΔ200, but less than in WT. Similarly, the K4me2 that would normally be seen further downstream is only able to get to the K4me1 stage. Because of space limitations, we don't completely restate the conclusions and model from Soares et al, but hope readers will go back and read that paper as well if they still don't understand the results.

7. The Swd2-F250A mutant reduces Spp1-Myc occupancy on PMA1. What is the effect on Set1 occupancy? How disruptive is this mutation to complex integrity in cells? This is important because the authors would like to argue that this mutation is having specific effects on COMPASS recruitment to RNA Pol II. If the entire COMPASS complex is disrupted, a reduction in H3K4me3 would not be as interesting. The co-IPs in Figure 5b show greatly reduced association of Spp1 and Set1 with the Swd2-F250A mutant. What about other subunits? It doesn't seem relevant to compare the co-IPs with the in vitro reconstitution. In the reconstituted complex the Swd2 mutation does not seem to affect Set1 and Spp1 incorporation in the complex. Also, the effects of the F250A substitution on H3K4me3 in vitro are subtle compared to effects in vivo. Fitting with the model, this could be due to the requirement for Swd2 in recruiting RNA Pol II to genes in vivo, but this was not clearly stated in that section of the Results. Overall, more caution is needed in interpreting this mutant unless the authors have additional data supporting its specificity.

Reviewer 1 also raised this point, so please see our answer above as to why we think the defects in the reconstituted complex are milder than in vivo. As we pointed out, we have done a new experiment IPing Set1 and looking for WT or F250A association, but this experiment is done in the presence of untagged Swd2 to bypass the feedback mechanism that had been causing degradation of Set1 in the Swd2 F250A. The reviewer had intuited correctly that we believe the F250A phenotypes results from weaker Set1 interactions that lead to inefficient COMPASS recruitment to genes. We have re-worded the paper to make our interpretations clearer and more cautious.

8. Does the Swd2-F250A mutation disrupt the Y2H interaction between Set1 and the CTD?

Great question. We now show in the new Fig. 5c that Swd2 F250A, much like the Swd2 deletion (Fig 3), strongly reduces the Set1-CTD Y2H signal.

9. Figure S1b: Molecular weight markers are needed.

These have been added.

10. Line 162: "isogeneicity" is misspelled.

Changed to "isogeneity".

11. Figure S2b: There are many bands in the Set1 lanes. On what basis was the band for full length Set1 assigned? Was the blot probed with anti-Set1 antibody?

This blot was probed with anti-Gal4 BD to detect the Set1-Gal4 fusions. We assigned the band in this blot based on alignment with Fig.S1c, where the fusion was detected with both anti-Gal4 BD and anti-Set1 antibodies. Full length Set1 is degraded rapidly in cells, so we assume many of the other smaller bands are breakdown products. We also note a strong cross-reacting band just under the size of full-length Set1 that is seen even in set1 deletion strains.

12. Figure S2c: Can the authors explain the H3K4me3 enrichment in the 3' UTR? Is there a noncoding transcript in this region?

This is due to the promoter of the adjacent gene SSP120, which we now include in the schematic of the locus.

Reviewer #3 (Remarks to the Author):

Review comments:

The study by Bae et. al. focused on characterization of the direct interaction between Pol II-CTD and subunits of the ySET1 complex. Despite the general assumption that ySET1 complex is directly recruited by Pol II to target genes, the exact molecular mechanism is not clear. The manuscript clearly demonstrated that N-terminal of ySET1 and Swd2 cooperatively interact with Pol II-CTD and this interaction is important for H3 K4 methylation in yeast. Furthermore, it showed that this n-SET/Swd2/Pol-II CTD regulation is independent of PAF1 and H2BK120ub and thus convincingly argued against a previous study that Swd2 functions through Bre1/Rad6 in H3K4me regulation. Most of the experiments were performed at high quality and with well-designed controls. There are only several weaknesses that need to be addressed before publication.

1. It is interesting that S Δ 200 and NS Δ 200 shift H3K4me2 to promoter proximal regions. Better explanation is warranted. Does this mean that these ySET1 mutants no longer interact with Pol II? This would argue against the proposed function by NS Δ 200.

Please see our response the similar questions from Reviewers 1 and 2.

2. Figure 4f, why NS Δ 200 had two bands?

As we showed in Soares 2014, Set1 levels are kept low by proteolysis, so in this blot and others in the paper we often see smaller bands that appear to be breakdown products. A similar band is also see with WT Set1 (lanes 3 and 4 of that blot).

3. Figure 4g, the pull-down assay seemed to suggest a much weaker interaction between CTD and NS Δ 200. However, rescue for H3K4me2/3 seemed much better. What is the reason for discrepancy?

Reviewer 1 also raised this point. We agree that the earlier figure did not present the results convincingly, so we re-ran the exact same samples, but loaded the lanes in a different order, separating SWD2 from the swd2 deletions. We believe the new panel 4g makes interpretation much easier. The quantitations below show that the level of partial rescue by NS Δ 200 is similar in both panels. Please see our reply to Reviewer 1 for the new panel and quantitation.

Because me2/3 is a result of multiple transcription cycles, changes in Set1 binding (measured by ChIP) are not necessarily going to be linear with higher methylation levels.

4. Figure 5 discussed a Swd2 mutant F250A, which was defective for Spp1 interaction. Since Spp1 is important for H3K4me regulation and indirectly interact with Pol II, it is not clear what exactly the function of this mutant. Is it possible that this mutant functions through spp1 instead of the Set1/Swd2/CTD axis? This has to be better clarified. If this mutant is indeed function through Spp1, what is the rationale to include it here?

We again apologize for apparently describing the Spp1 results in a confusing way. The new COMPASS structures (and earlier interaction studies) show that Spp1 interacts with the COMPASS catalytic core (WRAD) independently of Swd2, so the loss of Swd2-Spp1 colP seen in Fig 5b is indirectly due to the loss of Swd2-Set1 interaction. That's why we don't see a reduction of Spp1 in the complex reconstituted with Swd2 F250A, and think it's very unlikely that the Swd2 phenotypes arise indirectly through Spp1. Similar questions about this figure were also raised by Reviewers 1 and 2, so clearly we did not do a good job explaining the F250 experiments. We've tried to clarify in the revised paper. Due to the addition of new experiments, note that we moved this experiment to Sup Fig 5c, which now also includes a similar experiment, but IP'd via the Swd3 subunit, that produced the same result.

5. A recent cryo-EM study for the ySET1-NCP complex by Hsu et al., is available in bioRxiv. It elegantly demonstrates how H2BK120ub regulates H3K4me by removing the autoinhibitory ARM loop in n-SET. This should be cited and discussed. Given this new information, figure 6 is probably not necessary or at least be discussed in the context of the ySET1-NCP structure.

As we also told Reviewer 2, while our paper was being reviewed, we were very pleased to see the multiple structure papers showing how H2Bub interacts with the catalytic core subunits of COMPASS. We now incorporate this new information into our discussion. While these structures by themselves do not rule out additional interactions of H2Bub with Swd2, they are very consistent with our conclusions that bypassing Swd2 does not alleviate the requirement for H2Bub and PAF complex.

REVIEWERS' COMMENTS:

Reviewer #2 (Remarks to the Author):

The authors have done a nice job addressing previous reviewer concerns. The paper makes an important contribution to the chromatin and transcription fields by providing a molecular explanation for the recruitment of Set1 and the localization of H3 K4 methylation to active genes.

One minor comment:

Lines 150-152. The meaning of this sentence is unclear as written. An edit is needed.

Reviewer #3 (Remarks to the Author):

The authors have satisfactorily addressed all my previous concerns. The manuscript is acceptable for publication.

RESPONSE TO REVIEWERS' COMMENTS:

Reviewer #2 (Remarks to the Author):

The authors have done a nice job addressing previous reviewer concerns. The paper makes an important contribution to the chromatin and transcription fields by providing a molecular explanation for the recruitment of Set1 and the localization of H3 K4 methylation to active genes.

One minor comment:

Lines 150-152. The meaning of this sentence is unclear as written. An edit is needed.

This sentence (and paragraph) has been re-written to clarify our reasoning.

Reviewer #3 (Remarks to the Author):

The authors have satisfactorily addressed all my previous concerns. The manuscript is acceptable for publication.

We thank all the reviewers for their helpful comments.